# Effect of community health education on mothers' knowledge of obstetric danger signs and birth preparedness and complication readiness practices in southern Ethiopia: A cluster randomized controlled trial

Amanuel Yoseph[1]*, Wondwosen Teklesilasie[1], Francisco Guillen-Grima[2,3,4,5], Ayalew Astatkie[1]

1 School of Public Health, College of Medicine and Health Sciences, Hawassa University, Hawassa, Ethiopia, 2 Department of Health Sciences, Public University of Navarra, Pamplona, Spain, 3 Healthcare Research Institute of Navarra (IdiSNA), Pamplona, Spain, 4 Department of Preventive Medicine, Clinica Universidad de Navarra, Pamplona, Spain, 5 CIBER in Epidemiology and Public Health (CIBERESP), Institute of Health Carlos III, Madrid, Spain

* amanuelyoseph45@gmail.com

## Abstract

### Introduction

Increasing knowledge of obstetric danger signs (ODS) and encouraging birth preparedness and complication readiness (BPCR) practices are strategies to increase skilled maternal health service utilization in low-income countries. One of the methods to increase mothers' knowledge about ODS and promote BPCR practice is through health education intervention (HEI). However, the effect of context-specific community-based health education led by women's groups on these outcomes has yet to be comprehensively studied, and the existing evidence is inconclusive. Thus, we aimed to evaluate the effect of a context-specific community-based HEI led by women's groups on mothers' knowledge regarding ODS and BPCR practices in southern Ethiopia.

### Methods

An open-label, two-arm parallel group cluster-randomized controlled trial was conducted from January to August 2023 on pregnant women from 24 clusters (*kebeles*) (12 interventions and 12 controls) in the northern zone of the Sidama region. The Open Data Kit smartphone application was utilized to collect data. The intention-to-treat analysis was used to compare outcomes between groups. We fitted multilevel mixed-effects modified Poisson regression with robust standard error to account for between and within cluster effects.

### Results

One thousand and seventy pregnant women (540 in the intervention and 530 in the control clusters) responded to this study, making the overall response rate 95.02%. Excessive

**Data Availability Statement:** All relevant data are within the manuscript and its Supporting information files.

**Funding:** The authors received funding from Hawassa University and the Sidama Region President Office. The funders had no role in study design, data collection and analysis, decision to publish, or preparation of the manuscript.

**Competing interests:** The authors have declared that no competing interests exist.

**Abbreviations:** AIC, Akaike's Information Criterion; ANC, Antenatal Care; ARRs, Adjusted Risk Ratio; BIC, Bayesian Information Criterion; BPCR, Birth Preparedness and Complication Readiness; CI, Confidence Interval; cRCT, Cluster Randomized Controlled Trial; CRR, Crude Risk Ratio; EAs, Enumeration areas; GTP, Growth and Transformation Plan; HCG, Human Chorionic Gonadotropin; HCP, Health Care Provider; HEI, Health Education Intervention; HEW, Health Extension Worker; HF, Health Facility; HFD, Health facility Delivery; ICC, Intra-Cluster Correlation Coefficient; IRB, Institutional Review Board; ITA, Intention to Treat Analysis; MHSU, Maternal Health Service Utilization; ODK, Open Data Kit; ODS, Obstetric Danger Sign; PCA, Principal Component Analysis; PI, Principal Investigator; PNC, Postnatal Care; SBA, Skilled Birth Attendant; SD, Standard Deviation; VIF, Variance Inflation Factor; WDA, Women Development Army; WDT, Women Development Team; WHO, World Health Organization; WRA, Women of Reproductive Age.

vaginal bleeding (94.3% in the interventional group vs. 88.7% in the control group) was the commonest ODS mentioned during childbirth. Overall, 68.7% of women in the intervention group and 36.2% of mothers in the control group had good knowledge of ODS (P-value < 0.001). Saving money and materials (97.1% in the interventional group vs. 92.7% in the control group) was the most frequently practiced BPCR plan. Overall, 64.3% of women in the intervention group and 38.9% of mothers in the control group practiced BPCR (P-value < 0.001). HEI significantly increased overall knowledge of ODS (adjusted risk ratio [ARR]: 1.71; 99% CI: 1.14–2.57) and improved overall BPCR practice (ARR: 1.55; 99% CI: 1.02–2.39).

## Conclusions

A community-based HEI led by women's groups improved mothers' knowledge regarding ODS and BPCR practices in a rural setting in southern Ethiopia. Interventions designed to increase women's knowledge of ODS and improve BPCR practice must implement context-specific, community-based HEI that aligns with World Health Organization recommendations.

## Trial registration

NCT05865873.

## Introduction

Maternity is a normal process that causes several anatomical and physiological changes during the prenatal, intrapartum, and postpartum periods [1, 2]. Every period would be a positive experience in terms of assumptions, confirming that mothers and newborns attain their maximum potential for health and welfare [3, 4]. However, it may contain unexpected complications that expose the women and their fetuses to dangers and related morbidities and mortalities [5]. These complications are manifest as obstetric danger signs (ODS) [6–8].

According to the World Health Organization (WHO) 2022 report, direct obstetric complications are the leading cause of maternal death, with bleeding being the first cause globally and accounting for nearly 28% of all deaths [1, 2]. Studies also reported that hemorrhaging and eclampsia are the leading causes of maternal death in Africa [9, 10]. Nevertheless, many of these complications are preventable if the women are aware of them and are identified early, adequately treated, and managed [2]. However, most women have poor knowledge of ODS, particularly in developing countries [1].

The fundamental cause of maternal mortality in many low-income countries during prenatal, intrapartum, and postpartum periods is described in terms of the three critical delays model. These include delays in identifying life-threatening ODS and deciding to seek health care, delaying arrival at the health facility, and delaying receiving timely, sufficient, and effective care at the health facility [11]. Due to poor knowledge of ODS, women postpone seeking obstetric treatment, contributing to high maternal morbidity and mortality in developing countries [12, 13].

Studies from Ethiopia also reported that women had poor knowledge of ODS, which ranged between 15.5 and 48% [14–16], contributing to high maternal mortality in the country

(412 maternal deaths per 100,000 LBs) [17] and very high mortality in rural settings (1142 maternal deaths per 100,000 LBs) [18].

During pregnancy, pregnant women and their families must prepare to welcome a newborn and potentially overcome any unpredicted obstetric complications. This practice is known as the birth preparedness and complication readiness (BPCR) plan [6, 19]. The BPCR plan is a method to encourage the timely utilization of skilled care, particularly during childbirth, according to the notion that preparing for delivery decreases delays in accessing this care [20]. In developing countries, the proportion of women preparing for birth and its obstetric complications is low (35%) [21]. Only 32% of pregnant women in Ethiopia had a birth preparedness and complications readiness plan [22]. The figure is much inferior in southern Ethiopia (18.3%) [20].

The Ethiopian government has designed different strategies and initiatives to increase mothers' knowledge of ODS and BPCR plans, which is very important to promote women's health and improve their survival [23, 24]. Regardless of the Ethiopian government's strategies and initiatives, mothers' knowledge of ODS and the BPCR practice still needs to improve at the country level and is very low in rural settings [14, 15, 20, 22]. Hence, health education is one of the approaches to increasing a mother's knowledge of ODS and the practice of BPCR. It is a method focused on education and communication to develop the desired behavior change [25, 26]. However, the effect of a health education intervention on women's knowledge regarding ODS and the BPCR practice has yet to be comprehensively studied, and the evidence needs to be stronger in designing effective and efficient strategies. Studies conducted in Sokoto State, Nigeria [27], Korogwe district of rural Tanzania [28], Mundri East County, South Sudan [29], and Lagos State, Nigeria [30] showed positive effects of HEI. However, the strength of the evidence these studies delivered is weak due to epidemiological and statistical limitations. For example, one of the studies used a purposive sampling method, lacked randomization, and had inadequate power, as evidenced by a small sample size (only 70 women) [31]. Another study utilized a convenient sampling method with a small sample size (only 120 participants) and subsequent low power [32]. Besides, these studies were quasi-experimental and lacked one or more true experimental study elements, such as random allocation of subjects to study groups, which frequently led to confounding and made it difficult to establish causality. It also has lower internal validity because other variables may account for the results, and it is not easy to know if all confounding variables have been included [33].

Furthermore, more cluster-randomized controlled trial (cRCT) studies are needed to evaluate the effect of community-based HEIs on ODS knowledge and BPCR practice in developing countries, including Ethiopia. Using a cRCT permits the research to provide strong evidence of whether or not an HEI has the intended causal effect on outcomes. Therefore, we aimed to assess the effect of health education on mothers' knowledge of ODS and BPCR practice in southern Ethiopia. The research question to be answered by the present trial is, in pregnant women in the Sidama region of southern Ethiopia, how does a community-based health education intervention facilitated by women's groups compare to routine health education in improving knowledge of obstetric danger signs and increasing the practice of birth preparedness and complication readiness?

## Methods

### Study area

This study was conducted in the Northern Zone of Ethiopia's Sidama Region. Sidama Region was established on June 18, 2020. It is the country's second-smallest regional state by geographical area, after Harari, and the fifth-most populous [34]. It is situated in southern

Ethiopia and is divided into four zones, namely the Southern, Northern, Central, and Eastern zones, as well as one city administration [35]. The northern zone is 273 kilometers south of Addis Ababa. It is divided into eight districts and two town administrations. The zone contains 162 *kebeles* (Ethiopia's bottom administrative units). According to the Sidama Region Health Bureau, the zone has a population of 1.29 million people. Women of reproductive age (WRA) account for 23.3% of the population. The zone has 144 health posts, 36 health centers, one general hospital, and four primary hospitals. The zone's potential health service coverage by public HFs is 70% [36].

## Study design and population

A community-based, parallel-group, two-arm cRCT was implemented from January 10 to August 1, 2023, among pregnant women in the Northern Zone of Sidama Region, Ethiopia. In this study, *kebeles*, which are subsets of districts, were considered clusters. We included all pregnant women whose gestational age was less than or equal to 12 weeks and who lived in the zone for at least six months. Pregnant mothers who planned to change residence during the implementation of the intervention had not voluntarily provided consent or had severe illnesses were excluded from this study. Severe maternal illness in the context of this study includes severe chronic diseases, mental illness, and severe hyperemesis gravidarum that need strict hospital follow-up. We identified 1,126 pregnancies using the existing women's development team (WDT) and HEW structures. They surveyed all the eligible households to check for the presence of pregnant women in the household. Pregnant women were identified using a two-stage screening procedure. First, women were asked about the symptoms and signs of pregnancy. If women reported symptoms and signs of pregnancy, they were subjected to the second screening process using a urine human chorionic gonadotropin (HCG) test. A urine test was performed on women who had missed their menstrual cyclic period for six weeks or longer. Women were recruited for the study if the urine HCG test results were positive. This study was designed to detect and enroll pregnant women before 12 weeks of pregnancy via home-to-home visits. WDTs and HEWs visits confirmed the women's pregnancy, screened for eligibility, and obtained written informed consent before randomization. The recruitment period lasted from November 1st to December 31st, 2022. We reported this study based on the CONSORT 2010 statement: extension to cluster randomized trials guideline, and the filled-in checklist is provided as S1 File.

## Sample size calculation

The minimum required sample size was calculated using OpenEpi version 3.01 based on the following considerations. Due to the lack of a previous cRCT on the same topic, assumptions on the proportion of women with knowledge of ODS in the control and intervention arms were taken from a previous quasi-experimental study [31]. The proportion before the intervention was taken as the proportion in the control arm, and the proportion after the intervention was taken as the proportion in the intervention arm. Accordingly, P1 = 45.7% (proportion of women's knowledge regarding ODS during pregnancy) in the control group, and P2 = 62.9% (proportion of women's knowledge regarding ODS during pregnancy in the intervention arm) [31]. A confidence level of 95% and a power of 80% were also considered. Accordingly, the estimated effective sample size for individual-based randomization was 286 for both groups.

We used the cluster randomization method to assign study participants to the intervention and control arms due to the nature of the intervention, which is more applicable at the group level. This design decreases the intervention's spillover effect and provides logistical simplicity.

However, clustering in sample size calculation requires considering the effect of clustering and calculating a variance inflation factor (VIF) to increase the study's statistical power [37, 38]. The minimum number of clusters was computed by multiplying both groups' effective sample size and interclass correlation coefficient (ICC) factors [39]. We received the typical value of the ICC factor of 0.05 from the range of values (0.01 to 0.05) based on the recommendations [39–41].

Consequently, the minimum needed cluster number was 286*0.05 = 3.29 for both groups. Nevertheless, to maintain the cluster's sufficiency and adequate power [37, 38], 24 clusters were included in this study. The effective sample size was multiplied by a variance inflation factor of 1.55 to account for the effect of the cluster. The VIF was calculated assuming an equal cluster size of 12 study subjects from 24 clusters. We used an ICC value of 0.05 (VIF = 1+ [(n-1) ICC]), where 'n' is the average cluster size [39–41]. Thus, the final estimated sample size was 444 (222 in the intervention arm and 222 in the control arm).

We also calculated the minimum sample size needed to evaluate the effect of HEI on BPCR practice. In this case, P1 = 3.0% (proportion of women who practice BPCR in the control group) and P2 = 15.5% (proportion of women who practice BPCR in the intervention group) [42]. The level of confidence was taken to be 95% and power 80%. Based on these considerations, the final estimated sample size was 342 for both groups (171 for the intervention group and 171 for the control group). However, this study was part of a larger project, and the sample size calculated for another study, designed to evaluate the effect of the intervention on maternal health service utilization (MHSU), was 1,126 (i.e., more prominent). Hence, the sample size of 1,126 obtained for the other study was utilized as well, as it would suffice for both studies.

## Randomization

Randomization was done after securing consent and enrolling all study participants. The randomization was done by an autonomous, blinded statistician using an SPSS random number generator. Stratified by place of residence, *kebeles* were allocated by simple random assignment to either the HEI or control group. A cluster in our study was the *kebeles* (lowest administrative units in Ethiopia) of each district, which offered logistical simplicity and reduced the intervention spillover into the control arm. The study consisted of 24 *kebeles* from four randomly selected districts. These *kebeles* were stratified into two strata based on the place of residence: rural and urban *kebeles*. Stratification based on location reduces stratum variation and helps to balance the baseline covariates between the two arms [37, 38]. Also, it helps to balance the rural/urban disparity of maternal knowledge of ODS and BPCR practice. Similar clusters from each stratum were allocated to both groups to increase the resemblance between the two arms. Hence, three urban *kebeles* from the four districts were allocated to each group using an SPSS random number generator (in total, six *kebeles*). Similarly, nine rural *kebeles* from the four districts were allocated to each arm (in total, 18 *kebeles*). Finally, we included 47 pregnant women from each cluster.

## HEI procedure

WDT leaders who can read and write the *Sidaamu Afoo* language and are willing to do the intervention were recruited to deliver it. Following the recruitment, intensive training was given for three days on topics such as normal pregnancy and childbirth, ODS during pregnancy, delivery, and the postpartum period, the practice of BPCR, and maternal health service utilization (MHSU). The training also included intervention delivery strategies like properly handling pre-recorded audio material and posters, ethical considerations, and who to contact

with specific concerns related to the study intervention. The HEI was facilitated by WDT leaders at a small community meeting place using pre-recorded audio-based messages twice per month (detailed information provided in the S2 File).

## Study variables

For this study, the dependent variables were mothers' knowledge of ODS and BPCR practices. Each outcome variable has a binary outcome and was assessed using self-reported data from women. Maternal knowledge regarding ODS was measured using 30 questions during three phases, namely antepartum (9 questions), intrapartum (12 questions), and postpartum (9 questions). The correct answers were assigned a score of 1, while the incorrect answers were assigned a score of 0. The total knowledge scores range from 0 to 30. The study respondents who spontaneously mentioned at least 3 ODS during each phase were classified as having "good knowledge," and those who spontaneously mentioned two or fewer ODS were classified as having "poor knowledge" [43]. BPCR practice was measured using a questionnaire having five components as to whether or not the woman planned for her index pregnancy, such as identifying a closer proper HF for childbirth; finding and communicating an SBA; saving money; preparing material resources for childbirth and preparing for other associated costs; preparing or arranging transportation to a proper HF in case of childbirth and obstetric emergency; and identifying and fixing compatible blood group givers in case of blood requirements. If a woman prearranged at least two components out of 5, she was considered as having "well prepared" and otherwise considered "poorly prepared" [21, 44]. The intervention or exposure variable was health education. The intervention group received routine plus pre-recorded audio-based HEI augmented by posters in a small community meeting twice a month for six months, while the comparator group received the routine health education package until delivery as per the Ethiopian guideline [45]. Table 1 of S2 File contains information on how the variables were measured for this study.

## Blinding

Because of the nature of the intervention, neither the research team members nor the study participants could be blinded (open-label). However, the data collectors (outcome assessors) were masked or unaware of the subject's group allocation.

## Data collection procedures

We used a pre-tested and structured questionnaire to collect data (S3 File). It was adopted from earlier research of a similar nature [21, 43, 44]. The questionnaire was initially developed in English and translated into the *Sidaamu Afoo* language (primarily spoken in the study area). The questionnaire was translated back to English to ensure its consistency and originality. Two translators, English experts, and fluent *Sidaamu Afoo* speakers carried out the forward and backward translations. The translated questionnaire was reviewed by the principal investigator (PI) and a third individual who was likewise fluent in both languages. Then, based on the issues identified during the evaluation, any inaccuracy or inconsistency between the two versions was addressed.

The PI trained the data collectors and supervisors for two days before data collection on the significance of the study, data collection processes, aims, methodologies, and ethical considerations. Before data collection, the tool was pre-tested on 5% of the sample in the Dale district of the Sidama region and revised prior to actual data collection. The data were collected after seven weeks of delivery (end of the postnatal period). The health professionals with bachelor's degrees who were blinded about the intervention status collected the data through a face-to-

face interviewer-administered questionnaire at the women's homes utilizing the Open Data Kit (ODK) smartphone application.

To minimize the risk of bias, we exerted maximum efforts to maximize response and follow-up rates by intensively training data collectors and supervisors, masking outcomes assessors to the intervention assignment, and applying randomization. Daily, data was archived and uploaded to the Kobo Toolbox server.

## Data analysis technique

We reported absolute frequencies and percentages for categorical variables as summary measures. The mean with standard deviation (SD) was reported as a descriptive measure for numerical variables after the distribution was checked for normality. The wealth index was derived using principal component analysis (PCA) as a combined indicator of life standards based on 44 questions relating to ownership of carefully chosen household assets and basic amenities [46, 47]. Table 2 of S2 File describes this study's wealth index calculation process.

We used intention-to-treat analysis (ITA), which means we analyzed women included in the trial at the beginning and had the outcomes measured. We randomly allocated the intervention at the cluster level but evaluated the outcome at the individual level. A chi-square test was used to assess the effect of HEI on mothers' knowledge regarding ODS and BPCR practice in an unadjusted analysis.

A modified Poisson regression with robust standard error was used to calculate the risk ratios with 95% confidence intervals (CIs) for the effect of intervention on outcomes. We first performed a mixed effect-multilevel logistic regression with a random intercept model to determine whether a multilevel analysis was necessary. This model provides information on ICC, which is used to decide whether or not a multilevel model is necessary [48, 49]. The multilevel analysis model must be considered if the random intercept variance is significant or the ICC value exceeds 5%.

Four models were evaluated: Model 1 was the empty model, Model 2 included the intervention variable and other individual-level covariates, Model 3 contained only community-level covariates, and Model 4 contained the intervention variable and individual and community-level covariates. The random effect model was evaluated using the ICC value [50]. The ICC value was used to determine the proportion of variability in ODS knowledge and BPCR practice due to the clustering variable. The Akaike's information criterion (AIC), Bayesian information criterion (BIC), and log-likelihood with likelihood ratio test were used to determine which model best suited the data. The best-fitting model can be indicated by the lowest value of these criteria or a significant likelihood ratio test [51].

A multivariable regression model included the intervention variable, covariates with p-values < 0.25 on bi-variable analysis, and other covariates known to have practical significance with relevant support from the medical literature [52]. Effect modification was evaluated by entering interaction terms into the multivariable analysis model one at a time. Multicollinearity among the independent variables was also evaluated using a multiple linear regression model. We declared that the effect of multicollinearity would be less likely when the variance inflation factor was less than 5 for all variables [53].

Statistical significance was set at a p-value of < 0.05 for this analysis. This statistical significance level was adjusted to account for type I error inflation, which can result from the effect of multiple comparisons or testing problems in a single study. We adjusted it using the Bonferroni correction method. The adjusted significance level was computed by dividing the preset significance level by the number of statistical tests conducted (outcome variables). In our case, the adjusted level of significance was 0.05/5 = 0.01. Thus, a statistically significant association

was declared when the p-value was less than 0.01 [54, 55]. ARRs with 99% CIs were utilized to assess whether a statistically significant association existed and its strength. A statistically significant association between the exposures and outcome variables was validated when the 99% CIs of the ARRs did not contain 1. The complete Stata set of data on which this manuscript is based was provided in the S4 File.

## Ethics statement

This study received ethical approval from the institutional review board (IRB) at the College of Medicine and Health Sciences of Hawassa University, with reference number IRB/076/15. A support letter was obtained from the School of Public Health of Hawassa University, Sidama Region Health Bureau, district health offices, and *kebele* administrators.

Two levels of consent were obtained before conducting the actual study. First, community leaders approved on behalf of the community before randomization. Second, written informed consent was received from all study participants who met inclusion criteria before enrollment. Before signing informed written consent, study participants were informed about the purpose of the study, data collection techniques, voluntary participation, privacy, potential benefits, and dangers. All data collection methods and intervention techniques were carried out with confidentiality. After obtaining ethical approval, we registered the trial protocol at Clinical-Trials.gov with registration number NCT05865873 and trial protocol is provided in S5 File.

## Result

Fig 1 summarizes the details of the trial's randomization, recruitment, and eligibility procedures. Between November and December 2022, we assessed 1,440 pregnant mothers for eligibility; 1,126 from 24 clusters satisfied the criteria and were recruited for the study. WDT leaders and HEWs successfully communicated and recruited 1,126 eligible pregnant women from the 24 clusters (563 participants in 12 intervention clusters and 563 in 12 control clusters). A total of 1,070 (95.02%) women were available for outcome assessment during the data collection period: 540 in the intervention (95.91%) and 530 in the control (94.13%) groups. The study's overall response rate was 95.02%. The proportion of women lost to follow-up was comparable among both groups (4.98% in the intervention group vs. 5.87% in the control group). The mean (± SD) gestational age of women at recruitment was 10.72 (± 4.14) weeks.

Table 1 presents a summary of the socio-demographic and economic features of the participants. Most socio-demographic and economic features were comparable or well-balanced, with most respondents being of Sidama ethnicity, Protestant Christian region followers, married, and enrolled in primary school at the time of the interview. The mean (± SD) of the age of women was 29.21 (± 7.06) years in the intervention group and 28.76 (± 6.97) years in the control group. Four hundred ninety-six (91.9%) women in the intervention group and 478 (90.2%) women in the control group had attended formal education. Four hundred nine (77.2%) of the women in control and 386 (71.5%) in intervention groups were homemakers, whereas government employees constituted merely 7.7% in control and 13.1% in intervention groups. More than half of women, 306 (56.7%), in the intervention group and 231 (43.6%) in the control group had access to mass media like radio, television, and newspapers.

### Reproductive health characteristics

Most of the reproductive health characteristics were comparable between the intervention and control arms. The mean (± SD) of the age at first marriage of the women was 18.62 (± 0.97) years in the intervention group and 18.52 (± 1.21) years in the control group. Approximately one-tenth (10.9%) of the women in the intervention group and 12.6% in the control group had

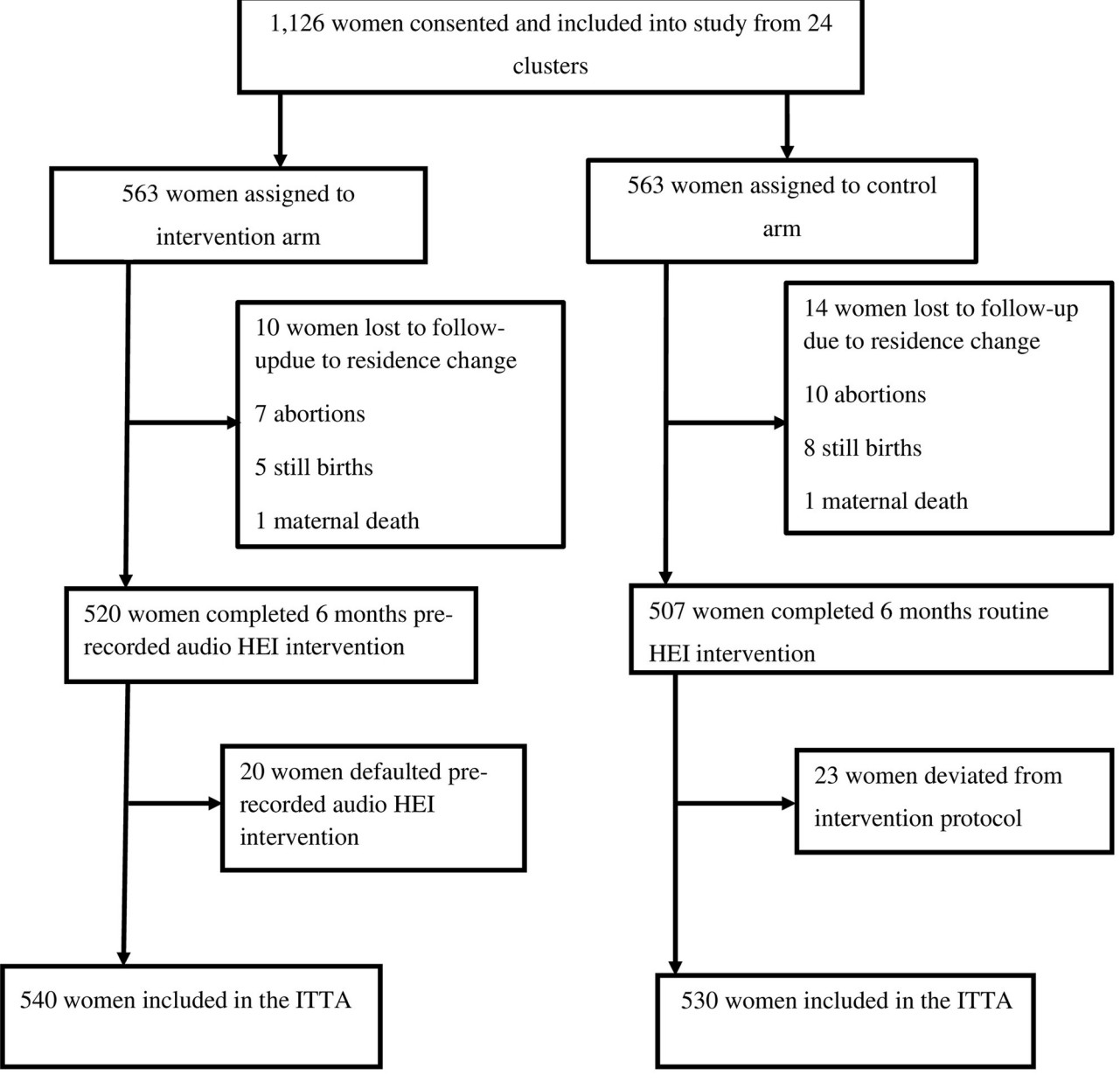

**Fig 1. Trial profile. Note**: HEI indicates health education intervention whereas ITTA indicates intention-to -treat analysis.

a previous history of abortion. Nearly half, 259 (48.9%), of the women in the intervention group had given birth to two to four children, compared to 277 (51.3%) women in the control group. Seven percent of study participants had experienced a stillbirth at least once in the intervention group, compared to nine percent in the control group. In more than two-thirds (65.1%) of the women in the control group, the last pregnancy was planned, compared to 81.5% in the interventional groups (Table 2).

## Description of the proportion of ODS knowledge and BPCR practice

Severe headaches (87.6% in the intervention group vs. 74.5% in the control group) and excessive vaginal bleeding (58.3% in the intervention group vs. 57.7% in the control group) were the

**Table 1. Socio-demographic characteristics of the trial participants (N = 1,070).**

| Variables | Intervention group | Control group | Total | P- value |
|---|---|---|---|---|
|  | N (%) | N (%) | N (%) |  |
| **Ethnicity** |  |  |  | 0.431 |
| Sidama | 485 (89.8) | 470 (88.7) | 955 (89.3) |  |
| Amhara | 23 (4.3) | 18 (3.4) | 41 (3.8) |  |
| Gurage | 12 (2.2) | 12 (2.3) | 24 (2.2) |  |
| Wolayita | 20 (3.7) | 30 (5.7) | 50 (4.7) |  |
| **Religions** |  |  |  | 0.311 |
| Protestant | 422 (78.1) | 391 (73.8) | 813 (76.0) |  |
| Orthodox | 45 (8.3) | 32 (6.0) | 77 (7.2) |  |
| Catholic | 38 (7.0) | 66 (12.5) | 104 (9.7) |  |
| Muslim | 35 (6.5) | 41 (7.7) | 76 (7.1) |  |
| **Mothers' education status** |  |  |  | 0.461 |
| Cannot read and write | 36 (6.7) | 39 (7.4) | 75 (7.0) |  |
| Can read and write only (without formal education) | 8 (1.5) | 13 (2.5) | 21 (2.0) |  |
| Have formal education | 496 (91.9) | 478 (90.2) | 974 (91.0) |  |
| **Women's occupation status** |  |  |  | 0.001 |
| Homemaker | 386 (71.5) | 409 (77.2) | 795 (74.3) |  |
| Farmer | 12 (2.2) | 37 (7.0) | 49 (4.6) |  |
| Government employee | 71 (13.1) | 41 (7.7) | 112 (10.5) |  |
| Merchant | 71 (13.1) | 43 (8.1) | 114 (10.7) |  |
| **Husband occupation status** |  |  |  | 0.001 |
| Government employee | 77 (14.3) | 40 (7.5) | 117 (10.9) |  |
| Merchant | 299 (55.4) | 247 (46.6) | 546 (51.0) |  |
| Farmer | 164 (30.4) | 243 (45.8) | 407 (38.0) |  |
| **Use of mass media** |  |  |  | 0.001 |
| No | 234 (43.3) | 299 (56.4) | 533 (49.8) |  |
| Yes | 306 (56.7) | 231 (43.6) | 537 (50.2) |  |
| **Wealth quintile** |  |  |  | 0.001 |
| Lowest | 131 (24.3) | 82 (15.5) | 213 (19.9) |  |
| Second | 77 (14.3) | 138 (26.0) | 215 (20.1) |  |
| Middle | 88 (16.3) | 126 (23.8) | 214 (20.0) |  |
| Fourth | 113 (20.9) | 101 (19.1) | 214 (20.0) |  |
| Highest | 131 (24.3) | 83 (15.7) | 214 (20.0) |  |

Note: The p-value in this table is based on the chi-square test.

most commonly mentioned ODS during pregnancy. Also, excessive vaginal bleeding (94.3% in the intervention group vs. 88.7% in the control group) was the most common ODS mentioned by women during childbirth, while prolapsed cord (11.5% in the intervention group vs. 4.7% in the control group) was the least mentioned. Likewise, excessive vaginal bleeding (93.7% in the intervention group vs. 86.2% in the control group) was the most common ODS mentioned by women during the postpartum period, while an inverted nipple (21.1% in the intervention group vs. 9.8% in the control group) was the least commonly mentioned ODS (Table 3 of S2 File). Overall, 68.7% of the mothers in the intervention group vs. 36.2% in the control group had good knowledge of ODS (P-value < 0.001) (Fig 2).

Saving money and material resources (97.1% in the intervention group vs. 92.7% in the control group) was the BPCR plan most commonly practiced by the women while identifying

**Table 2. Reproductive characteristics of the trial participants (N = 1,070).**

| Variables | Intervention group | Control group | Total | P- value |
|---|---|---|---|---|
| | N (%) | N (%) | N (%) | |
| **Previous history of abortions** | | | | |
| No | 481 (89.1) | 463 (87.4) | 944 (88.2) | 0.384 |
| Yes | 59 (10.9) | 67 (12.6) | 126 (11.8) | |
| **Total number of deliveries** | | | | |
| 1 | 182 (33.7) | 182 (34.3) | 364 (34.1) | 0.942 |
| 2–4 | 228 (42.2) | 225 (42.5) | 453 (42.3) | |
| ≥5 | 130 (24.1) | 123 (23.2) | 253 (23.6) | |
| **Previous history of neonatal death** | | | | 0.041 |
| No | 527 (97.6) | 505 (95.3) | 1032 (96.4) | |
| Yes | 13 (2.4) | 25 (4.7) | 38 (3.6) | |
| **Last pregnancy planned** | | | | 0.001 |
| No | 100 (18.5) | 185 (34.9) | 285 (26.6) | |
| Yes | 440 (81.5) | 345 (65.1) | 785 (73.4) | |
| **Encountered ODS during last pregnancy** | | | | 0.012 |
| No | 501 (92.8) | 468 (88.3) | 969 (90.6) | |
| Yes | 39 (7.2) | 62 (11.7) | 101 (9.4) | |
| **Faced ODS during last childbirth** | | | | 0.926 |
| No | 495 (91.7) | 485 (91.5) | 980 (91.6) | |
| Yes | 45 (8.3) | 45 (8.5) | 90 (8.4) | |
| **Confronted ODS during last postpartum period** | | | | 0.444 |
| No | 500 (92.6) | 497 (93.8) | 997 (93.2) | |
| Yes | 40 (7.4) | 33 (6.2) | 73 (6.8) | |

Note: The p-value in this table is based on the chi-square test for categorical data and the t-test for numeric data.

and fixing compatible blood group givers (6.9% in the intervention group vs. 5.6% in the control group) was the least commonly mentioned (Table 4 of S2 File). Overall, 64.3% of the mothers in the intervention group vs. 38.9% in the control group had BPCR practice (P-value < 0.001) (Fig 3).

## Effect of health education intervention on mothers' knowledge regarding ODS

In unadjusted analysis, the mothers' knowledge of ODS was significantly higher in the intervention group (68.7%) than in the control group (36.2%) (P-value < 0.001). After being adjusted for confounders and clustering, women who had received HEI had a 71% higher likelihood of ODS knowledge than women in the control group (ARR = 1.71; 99% CI: 1.14–2.57) (Table 3).

## Effect of health education intervention on BPCR practice

The BPCR practice was significantly different between the intervention group (64.3%) and the control group (38.9%) in unadjusted analysis (P-value < 0.001). After adjustment for confounders and clustering, women who had received HEI had 1.55 times higher likelihoods of BPCR practice than women in the control group (ARR = 1.55; 99% CI: 1.12–2.16) (Table 4).

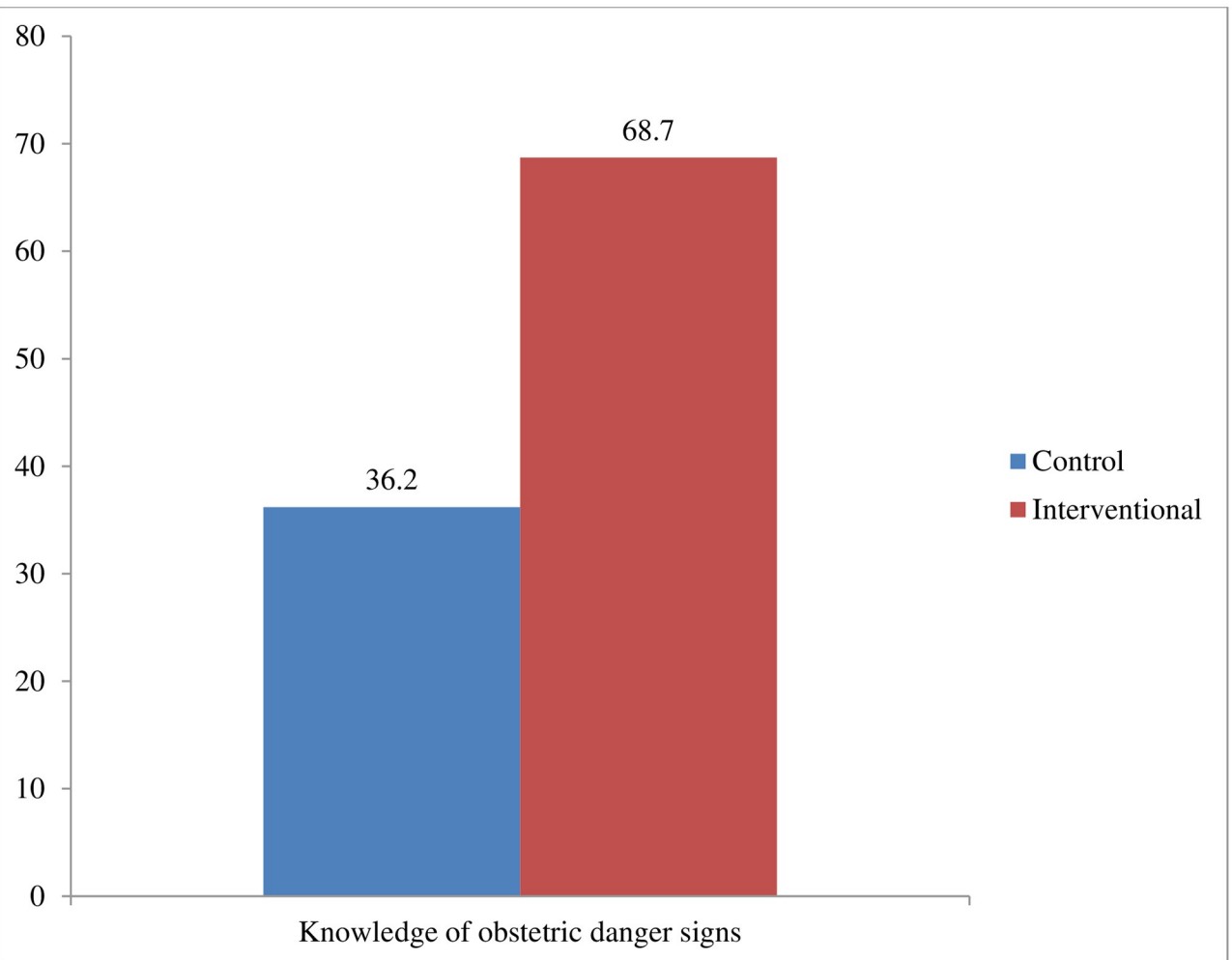

**Fig 2. Level of knowledge of obstetric danger signs among women in control and interventional arms.**

### Random effect model of ODS knowledge and BPCR practice

The multilevel mixed-effects modified Poisson regression with robust variance model fitted better than the ordinary model (p <0.001). The ICC value calculated using the intercept-only multilevel binary logistic model revealed that 27.46% of the variability in ODS knowledge and 38.78% in BPCR practice were related to membership in *kebeles* (Table 5 of S2 File).

### Model selection criteria

The model fitness evaluation test of ODS knowledge showed that the empty model was the least fit (AIC = 1801.97, BIC = 1811.92, and log-likelihood = -898.98). However, there was significant progress in the fitness of the models, particularly in the final model (AIC = 1775.71, BIC = 1790.15, and log-likelihood = -864.85). Thus, the final model is best fitted compared to the other models. Similarly, the model fitness significantly improved from the empty model to the final model in cases of BPCR practice (Table 5 of S2 File).

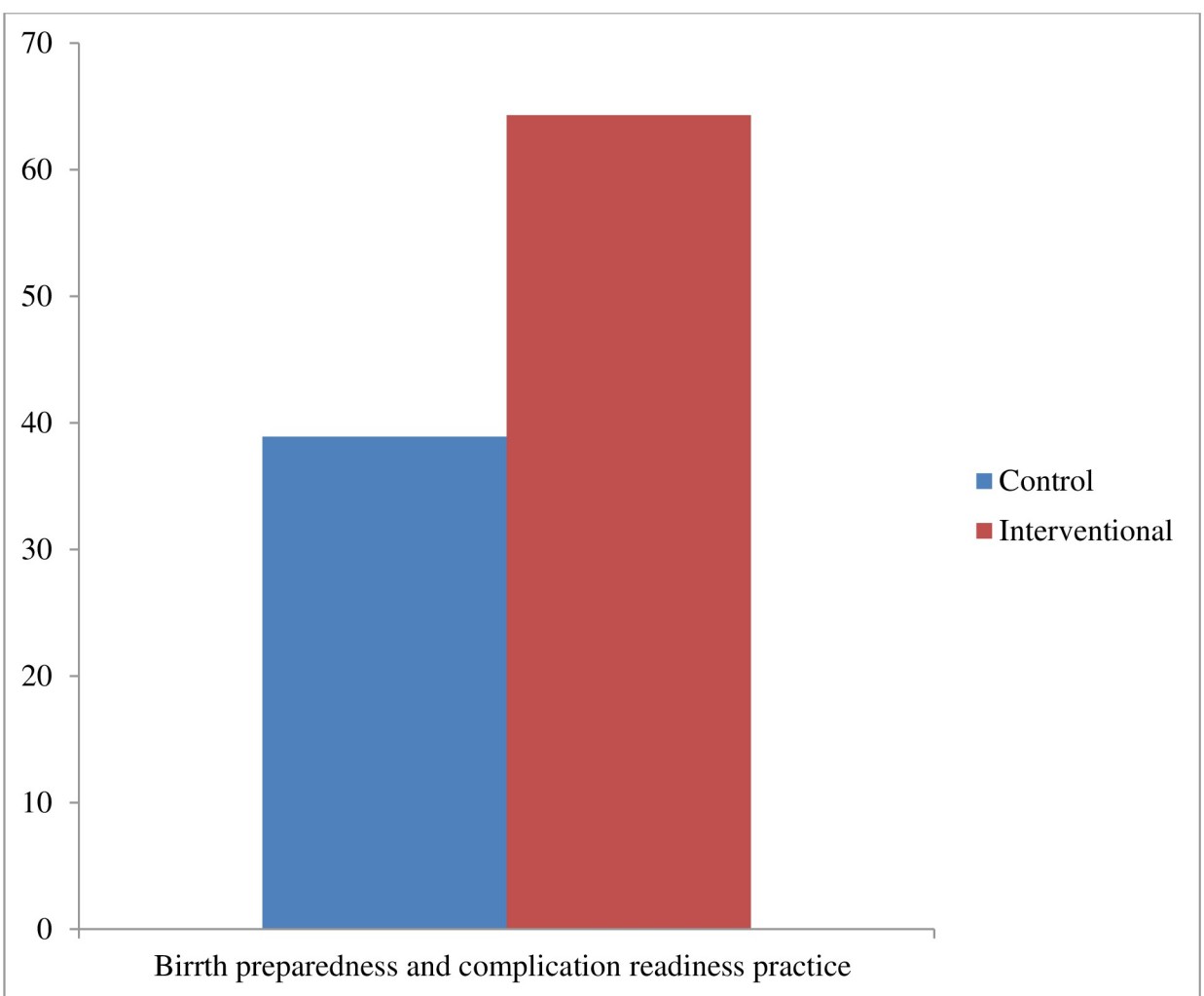

**Fig 3. Level of birth preparedness and complication readiness practice among women in control and interventional arms.**

**Table 3. Effect of health education intervention on knowledge of ODS among women of reproductive age in the Northern zone of the Sidama region, Ethiopia, 2023 (N = 1,070).**

| Variables | Knowledge of obstetric danger sign | | CRR (99% CI) | ARR (99% CI) |
|---|---|---|---|---|
| | **Good** | **Poor** | | |
| | **N (%)** | **N (%)** | | |
| **Study group** | | | | |
| Control | 192 (36.2) | 338 (63.8) | Ref | Ref |
| Intervention | 371 (68.7) | 169 (31.3) | 1.92 (1.37, 2.69) | 1.71 (1.14, 2.57)* |

Variables adjusted in the final model were women's occupation, husband's occupation, use of mass media, wealth quintile, previous history of neonatal death, last pregnancy planned, faced health problems during the pregnancy, road access, received model family training, availability of transport, place of residence, cluster-level mass media use, cluster-level distance, and cluster-level poverty.

*: significant association ($p < 0.01$); CI: confidence interval; ©: continuous variable; CRR: crude risk ratio; ARR: adjusted risk ratio; Ref: reference group.

**Table 4. Effect of health education intervention on BPCR practice among women of reproductive age in the Northern zone of the Sidama region, Ethiopia, 2023 (N = 1,070).**

| Variables | Birth preparedness and complication readiness | | CRR (99% CI) | ARR (99% CI) |
|---|---|---|---|---|
| | Poor | Well | | |
| | N (%) | N (%) | | |
| **Study group** | | | | |
| Control | 206 (38.9) | 324 (61.1) | Ref | Ref |
| Intervention | 347 (64.3) | 193 (35.7) | 1.70 (1.18, 2.44) | 1.55 (1.01, 2.39)* |

Variables adjusted in the final model were women's occupation, husband's occupation, use of mass media, wealth quintile, previous history of neonatal death, last pregnancy planned, faced health problems during the pregnancy, road access, received model family training, availability of transport, place of residence, cluster-level mass media use, cluster-level distance, and cluster-level poverty.

*: significant association ($p < 0.01$); CI: confidence interval; ©: continuous variable; CRR: crude risk ratio; ARR: adjusted risk ratio; Ref: reference group.

## Discussion

We evaluated the effect of community-based health education intervention facilitated by women's groups (women's development team) on mothers' knowledge regarding ODS and BPCR practice in the Sidama region of southern Ethiopia. Severe vaginal bleeding was the most common ODS mentioned by women during pregnancy, childbirth, and postpartum periods in both study arms. Saving money and materials was the most frequently mentioned BPCR practice in both study groups. The HEI significantly improved women's knowledge of ODS and BPCR practices.

The finding that severe vaginal bleeding was reported to be the most commonly mentioned ODS in both study arms during pregnancy, childbirth, and the postpartum period is similar to other findings from Ethiopia [56–60] and elsewhere [21, 59]. The similarity in findings might be because facility-based ANC counseling by HCPs often focuses on the commonest ODS like severe bleeding, reduced or absent fetal movement, and mal-presentation, which might increase the mother's recall of severe vaginal bleeding signs due to frequent information obtained from HCPs. [60]. The other reason might be that the community-based pregnant women forum also mainly focuses on a few common ODS during their meetings. HEWs allow the women to discuss the common ODS in their local setting [45]. Even if our HEI provided balanced information about all danger signs because all are equally important in our study, the most common one was severe vaginal bleeding, which might be an eco-effect from facility-based ANC counseling by HCPs and the community-based pregnant women forum.

Saving money and materials was the most frequently mentioned BPCR practice in both study groups. Similar findings were reported from earlier studies in the North Shewa zone [61], Northwestern Ethiopia [62], and rural Uganda [21]. In rural societies, it is customary to prepare money and materials such as towels, cups for newborns, butter, and porridge flour. Due to its ecological impact, this custom became commonplace in our study despite our equal emphasis on all components during HEI.

Community-based HEI increased the likelihood of women's knowledge regarding ODS. Consistent results were reported from the studies done in Sokoto State, Nigeria [27], rural Egypt [63], Lagos State, Nigeria [64], Zagazig University Hospitals [65], Nicaragua [66], and Cairo University Hospitals [67]. The reason could be that most earlier studies revealed that advanced levels of education were linked to enhanced cognitive skills, improved information processing abilities, and better values, which help to increase knowledge [68–70].

The current finding has a significant advantage because it considers the effects of community-based HEI as opposed to various intervention packages provided by those previously conducted studies [71–75]. The evidence in this study shows the effectiveness of HEI in increasing mothers' knowledge of ODS. The women's ability to identify ODS should minimize the delay in receiving basic emergency obstetric care, leading to higher rates of MHSU. However, there needs to be more evidence to comprehensively assess the effects of knowledge learned from HEI on MHSU. Thus, our studies in another paper evaluated whether this knowledge of ODS obtained from HEI might be changed into an increase in MHSU in a study setting similar to those observed following intervention in Nigeria [27], Egypt [63], and Nicaragua [66]. Thus, HEI could be a beneficial supplement to women's education in rural areas, increasing the utilization of MHS and preventing delays in emergency obstetric care. Promoting community-based health education in rural areas could be an effective way to increase MHSU in areas with low levels of education.

Community-based HEI increased the likelihood of women's BPCR practice. This result agreed with previous studies conducted in the Korogwe district of rural Tanzania [28], Zaria Metropolis [76], Mirzapur of Bangladesh [75], and Mundri East County, South Sudan [29]. This result agrees with the theory of reasoned action, which states that a person's intention determines whether they do a specific behavior. It is a function of one's attitude and the effect of the social environment, which can be positive or negative for a particular behavior [77, 78]. Thus, the women believe practicing BPCR will lead to a whole-positive outcome (i.e., HFD use). Then, they will hold a positive attitude toward executing that behavior (BPCR practice). Our result inferred that women's exposure to six months of community-based HEI has predisposed, modified, influenced, and changed their attitude toward BPCR practice. The other reason might be that well-prepared women have better knowledge of ODS and good communication with HCPs.

Consequently, they may have organized all the required prearrangements efficiently and effectively. Other researchers argued that women with good knowledge of ODS are more likely to be birth-prepared and have complications readiness practice [21, 79, 80]. Moreover, the community-based HEI might reach a broader audience of women and their families who may not access regular prenatal care at the HF level. The content of HEIs may also reach male listeners, who have an influential role in healthcare decision-making, and this may thus lead to an increase in BPCR practice.

It is relevant to note that community-based HEI significantly affects women's knowledge of ODS and BPCR practice among women in our study setting. To ensure the sustainability of knowledge obtained during HEI sessions and maintain the BPCR practice of women accessing MHS, the regional health bureau, and district office must reinforce women's health-seeking behaviors in the community of the study setting by using an already established structure. This reinforcement will help decrease maternal morbidity and mortality rates among women. The best way to deal with the first delays is to raise knowledge of ODS and BPCR practices with improved transportation networks. Meanwhile, their influence depends on the availability of high-quality health services at health facilities. Thus, focusing on both the demand and supply sides is crucial, even though they are only somewhat related (confidence in the quality of healthcare services increases demand) [73].

cRCT design is appropriate for interventions delivered at the group level and utilized when individual-level randomization is unlikely, or intervention is logically applicable to an entire group or naturally-existing clusters like schools, clinical practices, villages, *kebeles*, and enumeration areas (EAs) where the study subjects are school children, patients, villages, and inhabitants of *kebeles* [81, 82]. When allocating identifiable groups, cRCT is regarded as the strongest design in public health research intervention [83]. Based on the CONSORT 2010

statements, extension to cluster randomized trials, cRCTs are preferred when there is a risk of inadvertent spillover of intervention effects from one group to another (contamination) [84].

However, cross-contamination can happen in our study when women from one cluster contact women from others. In rural communities (relatives, friends, and neighbors), there are several chances for social mixing or interaction via migration or travel between control and intervention clusters. Likewise, there may be direct involvement or participation of dwellers from control clusters in intervention undertakings or, more probably, an informal dialogue of thoughts arising from intervention undertakings. Due to this, the control cluster dwellers may gain or obtain some basic information from hearing health messages provided for the intervention clusters. The most common problem with this type of cross-contamination may be the dilution of the effect of intervention differences between two arms [81, 82]. To prevent information cross-contamination between the two arms, we considered several measures. One such measure was the creation of a buffer zone, established by using at least four kebeles between the control and intervention clusters. The establishment was achieved using a map of each district. We assigned a midwife to clarify any issues and concerns from outside the study area regarding the HEI procedure during the intervention period. HEWs from a particular cluster made cRCT feasible and appropriate to prevent contamination. We faced some challenges during the implementation of the intervention components. During the early stages of pregnancy, due to cultural taboos, women were unwilling to notify their pregnancy status and underwent an HCG test. However, this has had no significant influence on our research findings.

The gold standard of randomized trials is determined by two qualities, namely randomization and double-blinding [85]. We could not blind study participants and the research team due to the nature of the intervention, but we masked data collectors (outcome assessors). However, this would not preclude the occurrence of bias, which might lead to an underestimation or overestimation of the intervention effect. Also, information bias could influence our results because the intervention was open-label, and the data were obtained from the women's self-report. Although difficult to quantify, women's knowledge of their exposure status (whether they received the intervention) likely influenced their self-reported responses to the knowledge and practice questions, thus leading to information bias. There is a potential for deliberately misreporting personally related variables like saving money and materials, preparing or arranging transportation to a proper HF in childbirth and obstetric emergencies, and identifying and fixing compatible blood group givers in case of blood requirements (social desirability bias). Therefore, the extent of these variables might have been overvalued, and as such, the association of the intervention with the magnitude of BPCR practice might have been overestimated. Regardless of these limitations, the results of this trial are sufficiently valid to develop appropriate intervention strategies and inform policy or program development.

Randomization is frequently assumed to eliminate selection bias and create similar groups regarding measurable and unmeasurable confounders. However, this assumption may not hold in cluster trials, especially when limited clusters are available. When a limited number of clusters are available, the risk of a baseline imbalance between the two arms can be significant [37, 38]. In this case, accounting for the effect of baseline covariates using multivariable analysis and assessing the covariates for intervention effect modification is frequently suggested [53, 85]. So, cognizant of this, we have accounted for the measured sources of confounders and assessed effect modification in our analysis. Most individual and community-level covariates were comparable between the intervention and control arms. However, some of them showed a significant imbalance between two arms like women's occupation, husband's occupation, mass media use, wealth index, women's pregnancy planned status, model family training, cluster-level distance to reach the nearest health facility, cluster-level mass media use, place of

residence, and cluster-level mass media, and these variables were adjusted for in the multivariable analysis. We also evaluated whether any of the covariates above modified the effect of the intervention. It was found that there was no significant effect modification because none of the interaction terms were statistically significant. Thus, our findings were not affected by modifying the covariates effect and were solely due to the intervention effect because we ruled out the internal validity threat [53].

The ICC value revealed that membership in *kebeles* explained 27.46% of the variability in ODS knowledge and 38.78% in BPCR practice. This result indicates that multilevel analysis should be considered because the ICC value is greater than 5%, which we considered [48, 49]. The units of analysis are treated as independent observations in traditional methods of ordinary regression. Regression coefficient standard errors will be underestimated if hierarchical structures are not recognized, which could result in an overestimation of statistical significance. Ignoring the clustering effect will likely affect the coefficients of higher-level determinants or standard errors. The effects of group-level determinants are confounded with the effects of group dummies in a fixed-effects ordinary model, so it is impossible to distinguish between effects due to unobserved and observed group characteristics. The effects of both types of variables can be estimated in a multilevel (random effects) model [49].

Another limitation involves the retention of women's knowledge of ODS. In this study, women's knowledge was measured immediately after the intervention. Future studies should measure the long-term retention of knowledge of ODS gained through HEIs at later intervals and with repeat exposure and the impact on the use of obstetric care services and maternal morbidity and mortality. We only had one follow-up period (i.e., six months), and we cannot conclude if knowledge and practice were maintained over more extended periods, especially in the intervention group. Also, the residual effect of the HEI needs to be evaluated via a postproject study after a few years of project completion to ensure persistent effects of the intervention in the study setting.

Moreover, we faced challenges in assessing the outcomes of all enrolled women due to various reasons, such as 24 mothers changing residence, 17 having abortions, 13 experiencing stillbirths, and 2 encountering maternal deaths. Consequently, there were missing outcome data, which is not in line with the principle of randomization. Randomization guarantees the comparability of two arms, meaning that they are balanced for HEI and unknown and known confounders only in the manner in which they were initially randomized. The missing women in the two arms can no longer be considered balanced when some members of either or both groups are eliminated. This bias might lead to an underestimation or overestimation of the intervention effect. Also, this situation decreases the sample size. It compromises the study's statistical power, making it unable to detect the true effect of the intervention or more susceptible to type II error (a high false negative result) [85]. However, this is minimal in our case because the percentage of women lost to follow-up in both groups was similar (4.98% in the intervention group vs. 5.87% in the control group). Also, we lost only 4.8% of randomized women, which is in line with the culture of less than 5% loss to follow-up, considered a low risk of bias in cRCTs that does not significantly affect ITTA results [85]. Moreover, we conducted a post-hoc analysis of power and obtained a statistical power of 100% for both ODS knowledge and BPCR practice, which is strong enough to detect intervention effects.

This trial has several strengths. From these, we registered the trial protocol at ClinicalTrials.gov with registration number NCT05865873 after obtaining ethical approval to avoid duplication. We used a cRCT study design comprising interventional and comparator groups to ascertain the temporal relationship, an essential epidemiological design to establish causality between intervention and outcome. Our sample size was large, which means it is adequate to identify the effects of HEI on outcomes. Hence, the findings are generalizable to all women of

reproductive age in study settings and vital to developing applicable policy strategies for efficient and effective promotion of women's knowledge of ODS and BPCR practice in the Sidama region and other parts of the country with similar contexts. A study from Sudan [29] and Tanzania [28] also found consistent results, suggesting that this inference may also apply to developing countries at comparable stages of socioeconomic development, culture, and health service access.

In the current study, women had significantly higher knowledge of ODS and BPCR practice after six months of HEI. In another paper, we comprehensively investigated the link between increased knowledge of ODS and BPCR practice and changes in women's behaviors that promote MHSU.

## Conclusions

The increased knowledge of ODS and BPCR practice among women in the study setting, resulting from community-based HEI, is particularly important for scaling the intervention to other regional, country, or similar settings. Therefore, our results support the WHO recommendation to include HEI in community-based health extension programs. Community-based HEI should be considered when planning interventions to increase women's knowledge of ODS and improve BPCR practice.

## Supporting information

**S1 File. CONSORT 2010 extension to cluster randomized controlled trial checklist.**
(DOC)

**S2 File. Some important information in the method and result section.**
(DOCX)

**S3 File. English and Sidamu *afoo* versions questionnaire.**
(DOCX)

**S4 File Stata data set.**
(DTA)

**S5 File. Study protocol.**
(DOCX)

## Acknowledgments

Our profound gratitude goes to former Sidama Media Network manager Mr. Birhanu Hankara and Ms. Selamawit Tibo, a media expert, for their cooperation and facilitation of recording high-quality HEI audio material. We also thank Ms. Mihrete Sunura for her excellent narration of the HEI messages in a language acceptable to the community and cultural context and her immense commitment and response to all phone calls from HEWs during the intervention period. Further, we thank all HEWs for their tremendous assistance during the study. From the bottom of our hearts, we want to thank Mr. Misale Jilo for his assistance in translating the research tools and the health education message, audio material development facilitation, and data collection supervision. We are also grateful to the study participants, supervisors, data collectors, and administrators at various levels in the Sidama Region who contributed directly and indirectly to this study. Finally, our greatest thanks go to Netsanet Kibru for her support in printing posters and funding portable Bluetooth devices (Gepps's).

## Author Contributions

**Conceptualization:** Amanuel Yoseph, Ayalew Astatkie.

**Data curation:** Amanuel Yoseph, Ayalew Astatkie.

**Formal analysis:** Amanuel Yoseph, Ayalew Astatkie.

**Funding acquisition:** Amanuel Yoseph, Ayalew Astatkie.

**Investigation:** Amanuel Yoseph, Wondwosen Teklesilasie, Francisco Guillen-Grima, Ayalew Astatkie.

**Methodology:** Amanuel Yoseph, Wondwosen Teklesilasie, Francisco Guillen-Grima, Ayalew Astatkie.

**Project administration:** Amanuel Yoseph, Wondwosen Teklesilasie, Francisco Guillen-Grima, Ayalew Astatkie.

**Resources:** Amanuel Yoseph, Ayalew Astatkie.

**Software:** Amanuel Yoseph, Ayalew Astatkie.

**Supervision:** Amanuel Yoseph, Ayalew Astatkie.

**Validation:** Amanuel Yoseph, Wondwosen Teklesilasie, Francisco Guillen-Grima, Ayalew Astatkie.

**Visualization:** Amanuel Yoseph, Wondwosen Teklesilasie, Francisco Guillen-Grima, Ayalew Astatkie.

**Writing – original draft:** Amanuel Yoseph, Wondwosen Teklesilasie, Francisco Guillen-Grima, Ayalew Astatkie.

**Writing – review & editing:** Amanuel Yoseph, Wondwosen Teklesilasie, Francisco Guillen-Grima, Ayalew Astatkie.

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
