## [Decision Letter · Decision Letter 0]

31 Jul 2024

PONE-D-23-40070Effect of community-based health education led by women's groups on mothers' knowledge of obstetric danger signs and birth preparedness and complication readiness practices in southern Ethiopia: A cluster randomized controlled trial.

PLOS ONE

Dear Dr. Yoseph Samago,

Thank you for submitting your manuscript to PLOS ONE. After careful consideration, we feel that it has merit but does not fully meet PLOS ONE’s publication criteria as it currently stands. Therefore, we invite you to submit a revised version of the manuscript that addresses the points raised during the review process.

 I currently have four reviews for your paper PONE-D-23-40070. The reviewers, who have important expertise in areas your paper covers, including health interventions and methodology, overall found the manuscript of potential interest and useful findings, however they raised important topics that need be addressed by author in the resubmission..

Taking into account the reviewers' comments will improve the manuscript for acceptance. The changes should be observed mainly in terms of synthesizing the introduction, strengthening the methodology and strengthening the discussion and conclusions.

My suggestions: follow the criteria for publications in PLOS ONE, reduce the number of keywords, and check consistency between all sections of the manuscript.

We look forward to receiving your revised manuscript.

Kind regards,

Doris Verónica Ortega-Altamirano, PhD

Academic Editor

PLOS ONE

 [Hawassa University and Sidama region president office].  

Please respond by return e-mail so that we can amend your financial disclosure and competing interests on your behalf.

Reviewers' comments:

Reviewer's Responses to Questions

**Comments to the Author**

1. Is the manuscript technically sound, and do the data support the conclusions?

Reviewer #1: Yes

2. Has the statistical analysis been performed appropriately and rigorously? 

Reviewer #1: Yes

3. Have the authors made all data underlying the findings in their manuscript fully available?

Reviewer #1: Yes

4. Is the manuscript presented in an intelligible fashion and written in standard English?

Reviewer #1: Yes

5. Review Comments to the Author

Reviewer #1: 1. There are a lot Keywords, usually require about 3-8 keywords but 9 are a lot.

2. The title is so long too. I suggest: "community health education on mothers' knowledge

with obstetric risk signs, birth preparedness and complication readiness practices in

southern Ethiopia: A cluster randomized controlled trial"

3. Focus on the variables identified in the title so as not to make the writing so extensive.

6. PLOS authors have the option to publish the peer review history of their article (what does this mean?). If published, this will include your full peer review and any attached files.

Reviewer #1: **Yes: **Claudia MACIAS-CARRILLO

---

## [Author Response · Author response to Decision Letter 0]

6 Aug 2024

Point-by-point responses to reviewers’ and editor’s comments

Academic editor

Comment 1: I currently have four reviews for your paper PONE-D-23-40070. The reviewers, who have important expertise in areas your paper covers, including health interventions and methodology, overall found the manuscript of potential interest and useful findings; however they raised important topics that need be addressed by author in the resubmission. Taking into account the reviewers' comments will improve the manuscript for acceptance. The changes should be observed mainly in terms of synthesizing the introduction, strengthening the methodology and strengthening the discussion and conclusions.

Authors’ response: Thank you for your kind remark. We duly considered all comments and have revised the manuscript accordingly.

Comment 2: My suggestions: follow the criteria for publications in PLOS ONE, reduce the number of keywords, and check consistency between all sections of the manuscript.

Authors’ response: Thank you a lot for this valuable comment. We have accepted the comment and made the required revision. The keywords have been edited to be consistent with PLOS ONE publications criteria as per your suggestion. Besides, we have checked the consistency of all section of our manuscript. 

Comment 3: Authors’ response: Thank you for this important notice. However, we didn’t change our financial disclosure because we have correctly stated it in the online submission system. Besides, we have already followed the figures files preparation style of PLOS ONE journal. 

Comment 4: If applicable, we recommend that you deposit your laboratory protocols in protocols.io to enhance the reproducibility of your results. Protocols.io assigns your protocol its own identifier (DOI) so that it can be cited independently in the future. 

Authors’ response: Thank you for this recommendation. This study hasn’t laboratory protocol but has a study protocol. We have already provided the study protocol as a supplementary material. 

Reviewer 1

Comment 1: There are a lot Keywords, usually require about 3-8 keywords but 9 are a lot.

Authors’ response: Thank you very much for your vital comments. As per your comment and comment 2 from the editor, we have revised the keywords to align with journal requirement. 

Comment 2: The title is so long too. I suggest: "community health education on mothers' knowledge with obstetric risk signs, birth preparedness and complication readiness practices in southern Ethiopia: A cluster randomized controlled trial"

Authors’ response: Thank you, too, for this comment. We have accepted your comment and made the required revisions as per your comment. The slight deviation in the title from your suggested title is meant only for grammatical correctness and clarity.

Comment 3: Focus on the variables identified in the title so as not to make the writing so extensive.

Authors’ response: Thank you for pointing this out. Yes, what you pointed out is a possibility. However, the clear definition of the exposures and outcomes are very important to provide comprehensive information for readers on how variables are measured. Moreover, the variables measurement is not accurately and consistently provided across the medical literature that precludes the replication of measurement and comparison of findings by scholar in different setting and time. Thus, we provided the clear definition separately to each variable to facilitate replication and comparison by different researchers. 

Journal requirements

Requirement 1: Please ensure that your manuscript meets PLOS ONE's style requirements, including those for file naming.

Authors’ response: We have followed PLOS ONE’s style requirements, including file naming conventions, in preparing the manuscript.

Requirement 2: Note from Emily Chenette, Editor in Chief of PLOS ONE, and Iain Hrynaszkiewicz, Director of Open Research Solutions at PLOS: Did you know that depositing data in a repository is associated with up to a 25% citation advantage (https://doi.org/10.1371/journal.pone.0230416)? If you’ve not already done so, consider depositing your raw data in a repository to ensure your work is read, appreciated and cited by the largest possible audience. You’ll also earn an Accessible Data icon on your published paper if you deposit your data in any participating repository (https://plos.org/open-science/open-data/#accessible-data).

Authors’ response: Thank you very much for this vital information. Now we have provided the full data set on repository in which the manuscript is prepared to take advantage of citation as per your suggestion. 

Requirement 3: Thank you for stating the following financial disclosure: 

 [Hawassa University and Sidama region president office]. Please state what role the funders took in the study. If the funders had no role, please state: "The funders had no role in study design, data collection and analysis, decision to publish, or preparation of the manuscript." If this statement is not correct you must amend it as needed. 

Please respond by return e-mail so that we can amend your financial disclosure and competing interests on your behalf.

Authors’ response: As we also responded to the above comment (comment 3), the financial disclosure statement has been correctly stated as per your suggestion during first online submission in the system. All the required details are provided.

---

## [Editor Report · Decision Letter 1]

15 Aug 2024

PONE-D-23-40070R1Effect of community health education on mothers' knowledge of obstetric danger signs and birth preparedness and complication readiness practices in southern Ethiopia: A cluster randomized controlled trialPLOS ONE

Dear Dr. Yoseph ,

Thank you for submitting your manuscript to PLOS ONE. After careful consideration, we feel that it has merit but does not fully meet PLOS ONE’s publication criteria as it currently stands. Therefore, we invite you to submit a revised version of the manuscript that addresses the points raised during the review process.

The comments and suggestions of 2-4 reviewers were not answered in their previous submission. I realized that it was a mistake and did not arrive properly. I apologize for the inconvenience. The decision on the Major Revision is still valid and may be answered by the authors in the following submission.

We look forward to receiving your revised manuscript.

Kind regards,

Doris Verónica Ortega-Altamirano, PhD

Academic Editor

PLOS ONE

Additional Editor Comments:

The comments and suggestions of 2-4 reviewers were not answered in their previous submission (R1). I realized that it was a mistake and did not arrive properly. I apologize for the inconvenience.

Reviewer 1

1. There are a lot Keywords, usually require about 3-8 keywords but 9 are a lot.

2. The title is so long too. I suggest: "community health education on mothers' knowledge

with obstetric risk signs, birth preparedness and complication readiness practices in

southern Ethiopia: A cluster randomized controlled trial"

3. Focus on the variables identified in the title so as not to make the writing so extensive.

Reviewer 2

Comments to authors:

Thank you for the opportunity to review this manuscript. Their results highlight the effectiveness of community-based interventions to increase women’s knowledge about pregnancy complications and best practices which is necessary to reduce maternal and infant mortality rates worldwide. The manuscript would benefit from some editing and summarizing some sections as it is very long and hard to follow at times. Please see comments below.

Introduction

1. The introduction could be significantly shortened, and some of the information used more in the discussion section. I would suggest the following structure;

a. Paragraph describing ODS including some prevalence data

b. Paragraph addressing the women’s knowledge gap of ODS

c. Paragraph describing interventions and government efforts

d. Paragraph describing the lack of research and objectives of this manuscript

2. It would be good for the authors to add some data on prevalence of ODS and maternal mortality particularly in Ethiopia.

Methods

3. Adding a flow chart or diagram depicting the participant’s selection and randomization process would be helpful to follow the study design, population and randomization paragraphs a little better.

4. Study variables: I suggest the authors only focus on variables that are relevant for this manuscript.

5. I suggest moving the description of the intervention to supplementary material and including only a brief summary of it in the manuscript. I also think it should be placed before talking about the outcomes.

6. Page 14 paragraph 2 talks about data collection and then goes back to randomization, this is a little confusing. I would suggest authors to mention everything that has to do with randomization in the appropriate paragraph and not coming both.

7. While the authors offer very detailed information and justifications for their model selection, I’m a little confused if their multilevel models were linear or Poisson models. It is not the standard to estimate ICC from a Poisson model and usually variance portioning is used. If fitting a logistic model then MOR is what is usually estimated as opposed to ICC If authors estimated ICC from non-linear models I would suggest that they add the formula of how they did it to the manuscript.

8. Why did authors decide the cutoff of p < 0.25 to be included in multivariate models

9. What was the reasoning behind the effect modification tests? Which variables were tested? I would suggest authors to elaborate more on this.

10. In general, authors provide many statistical terms and details to the methods section that makes it hard to follow. I would suggest authors simplify it and add only the important information readers would need in order to replicate this analysis. The section is very hard to follow and all the justification and details can be distracting.

Results

11. On the description of table 1, it would be important to highlight the differences between groups, given that this is an RCT and Table 1 should be used to assess balance between groups. The statistically significant differences for Mass Media access and wealth index are important and could be associated with the outcome as well so should be highlighted.

12. Similar to previous comment, it is important to highlight significant differences between groups, previous history of neonatal death and ODS during last pregnancy could be associated with the outcomes as well and should be highlighted as potential confounders.

13. I find tables 3 and 4 a little confusing. I would separate the N(%) of good and poor knowledge into a separate table and just leave the model results. Also, it is not very clear which models the authors are showing. In the methods they refer to a sequence of four models and they only show results for 2, clarifying that would be very helpful.

Discussion

14. While the information provided is good and authors do a good job of comparing their findings to other literature, the limitations section is very long and disorganized. I would strongly advise to summarize better the limitations with regards to clustered RCTs and then they can talk about limitations of analysis or other types instead of going back an forth which makes it hard to follow.

Reviewer 3

Amanuel Yoseph and co-authors evaluated the effect of community-based health education intervention facilitated by women's groups (women's development team) on mothers' knowledge of SDG and BPCR practice in Sidama region in the southern Ethiopia.

I would like to thank the authors for this excellent manuscript. A problem that affects a high proportion of women in underdeveloped countries is outlined: high maternal morbidity and mortality. But, even more importantly, the authors present evidence on effective strategies to mitigate this important public health problem. Adding to the relevance of the topic is the correct design, the careful handling of ethical aspects, a robust analysis and the writing of the manuscript.

My only suggestion is to review the title, its length seems a bit excessive to me. Additionally, I consider the comment "while another paper is focused on three skilled [...], in the "Study variables" section, unnecessary.

Congratulations, I really enjoyed reading the manuscript

Reviewer 4

The topic is relevant because of the social impact and inequalities that occur in developing countries. It is essential to recognize indicators of potential risks during childbirth in order to make informed decisions and prevent complications that could lead to the death of both mother and baby. However, we have some observations.

Overall, the current version of the article is quite dense and difficult to understand because of long and ambiguous sections. To ensure clarity for the reader, it is essential to reorganize the structure of the paper. In addition, it is substantial in the paper, to balance the section statistical analysis with importance of the impact of the intervention, will allowing the validity of the study to be measured.

The specific comments below recommend reducing the text in certain sections:

Introduction Section: From our perspective, the ideas in this section are long and repetitive. We recommend condensing sentences 2, 3, and 8. Sentence 6 could be rewritten or omitted.

In order to develop an educational intervention that addresses complications and reduces maternal and infant mortality rate, it is crucial to have a solid theoretical and causal framework. The causal framework is a diagram that considers the factors associated with social knowledge and establishes clear theoretical relationships between these factors. By incorporating a causal framework, the intervention can show its function and impact, highlighting the desired variable of change and the desired health outcomes. In addition, the baseline measure allows identifying should identify the variables that should be included in the analysis conceptual and ensure factor similarity between the intervention and non-intervention groups. It also allows for including relevant variables in the statistical analysis. The causal framework will justify the importance and scope of the intervention. For further help, see the following references:

• Rossi, P. H., Lipsey, M. W., & Freeman, H. E. (2004). Evaluation: A Systematic Approach (7th ed.). Thousand Oaks, CA: SAGE Publications. This classic program evaluation text underscores the importance of understanding and documenting causal relationships in intervention programs.

Centers for Disease Control and Prevention (CDC). (2011). Developing an Effective Evaluation Plan. Atlanta, GA: CDC, National Center for Chronic Disease Prevention and Health Promotion. This paper details the importance of causal and logic frameworks in health program evaluation.

• Funnell, S. C., & Rogers, P. J. (2011). Purposeful Program Theory: Effective Use of Theories of Change and Logic Models. San Francisco, CA: Jossey-Bass. This book provides comprehensive guidance on how to develop and use theories of change and logic models in program evaluation.

• Weiss, C. H. (1997). Theory-Based Evaluation: Past, Present, and Future. New Directions for Evaluation, 1997(76), 41-55. This article reviews the history and utility of theory-based evaluations, including the importance of causal frameworks.

• Bamberger, M., Rugh, J., & Mabry, L. (2012). RealWorld Evaluation: Working Under Budget, Time, Data, and Political Constraints (2nd ed.). Thousand Oaks, CA: SAGE Publications. This book addresses the challenges and strategies for evaluation in real-world settings, including justifying interventions through causal frameworks.

Method Section:

As a general comment, this section should be revised and summarized.

It is recommended that the participant recruitment process be described in more detail both in the text and in Figure 1.

I am confused by the way participant recruitment is described. If pregnant women were selected by visiting all households, if so, how was the response rate assessed if some households had a pregnant woman who did not want to be interviewed, or was not home because she had been hospitalized for complications of pregnancy? Other questions: How many households were visited in total? What is the rate of refusal or nonparticipation? During what stage of the recruitment process must women provide consent to take part? What was the community consent process? Given that the intervention was part of a major project, at what point in the study process were the groups randomized?

How do you measure the equivalence of the two groups in personal, social, and health service variables that theoretically take part in knowledge?

In the sample size calculation section, if a household census was conducted and pregnant women were identified, the most appropriate thing to do is to identify the statistical power or precision got when identifying knowledge in the total number of women who took part in the study.

On the other hand, what were the criteria for generating the clusters?

It would be appropriate to summarize this section as well.

After randomization: Are the factors that determine knowledge the same in both groups (intervened and non-intervened)? How do you check if randomization worked in creating kebeles?

In the variables section of the study, when it is noted that “The respondents in the study spontaneously mentioned three questions… spontaneous knowledge was defined… they could spontaneously name two or more…” Question: During the validation of the questionnaire and in particular of the knowledge questions, was the potential information bias that could influence the results quantified?, and in this context, did they measure its potential impact on the misclassification of knowledge in the statistical analysis?

Regarding the measurement of conceptual constructs such as “knowledge” during the analysis of the information, does the questions have the same value to measure the dimensions of the Knowledge construct?

In the last sentence of the same section, they show that this study is part of another larger project with another aim. It is not clear if it affected the creation of representativeness of women in the community or the recruitment and randomization process. Can you clarify?

IES process section. How did the topics offered to improve knowledge used in the intervention emerge? What competencies should participants have at the end of the educational training? Were assessments conducted before the intervention? What were the assessment results at the beginning and during the training process? If so, is there a parameter at the beginning of the intervention to measure the difference in knowledge between the intervention and non-intervention groups? If so, were these controlled for in the statistical analysis? Please comment.

Regarding the sections on data collection procedures, data techniques, and ethical statements, please reduce the length of these sections and rewrite them.

Results section. The statistical analysis should begin by evaluating the difference in the knowledge indicators that allow the initial counterfactual to be evaluated and induce the final counterfactual. It is important, measure the change before and after the intervention in both groups. A knowledge index be developed to meet the stated aim.

The tables are extensive. Please include the results of the important variables.

Explain the statistical differences that allow the identification of variables used in the adjusted statistical model.

Tables 3 and 4, please include at the bottom of the table the variables used to adjust the final model, and show the most essential findings that allow the subsequent discussion of the results.

Create a bar graph showing the percentage of increase in knowledge caused by the intervention that is the objective of the study.

In the Discussion section, it is recommended that the authors include the percentage of change in knowledge about the SDG and BPCR practices got at the end of the intervention.

An important aspect of the discussion of the study findings is the statistical analysis, which includes both intention to treat and final analysis.

The authors refer to an increased likelihood of women performing BPCR after the intervention, but do not specify which specific outcomes they are referring to.

The discussion mentions potential cost and material savings, but it is unclear whether this was one of the study’s goals.

In conclusion, the paper cannot be published as it is, observations and comments are necessary.

---

## [Author Response · Author response to Decision Letter 1]

28 Sep 2024

Point-by-point responses to reviewers’ and editor’s comments

Reviewer 1

Comment 1: There are a lot Keywords, usually require about 3-8 keywords but 9 are a lot.

Authors’ response: Thank you very much for your vital comments. As per your comment and comment 2 from editor, we have revised the keywords to align with journal requirement. 

Comment 2: The title is so long too. I suggest: "community health education on mothers' knowledge with obstetric risk signs, birth preparedness and complication readiness practices in southern Ethiopia: A cluster randomized controlled trial"

Authors’ response: Thank you, too, for this comment. We have accepted your comment and made the required revisions as per your comment. 

Comment 3: Focus on the variables identified in the title so as not to make the writing so extensive.

Authors’ response: Thank you for pointing this out. This comment is also shared by reviewers 2 and 4; we have revised this section of the manuscript as per the suggestions of the reviewers.

Reviewer 2

Comment 1: The introduction could be significantly shortened, and some of the information used more in the discussion section. I would suggest the following structure; a. Paragraph describing ODS including some prevalence data b. Paragraph addressing the women’s knowledge gap of ODS c. Paragraph describing interventions and government efforts d. Paragraph describing the lack of research and objectives of this manuscript 2. It would be good for the authors to add some data on prevalence of ODS and maternal mortality particularly in Ethiopia. 

Authors’ response: Thank you a lot for this valuable comment. We have accepted the comment and made the required revision. The introduction has been edited to be more logical and coherent by including prevalence of ODS and maternal mortality data as per your suggestion.

Comment 2: Adding a flow chart or diagram depicting the participant’s selection and randomization process would be helpful to follow the study design, population and randomization paragraphs a little better. 

Authors’ response: Dear reviewer, thank you for your critical comment. What you raised is very important to improve this section of the manuscript. However, the flow chart or diagram is already published in another paper (see figure 2). We haven’t provided the figure because to avoid dual publication of the same figure that can lead to scientific misconduct. See the link to the publication: https://doi.org/10.3390/healthcare12101045. 

Comment 3: Study variables: I suggest the authors only focus on variables that are relevant for this manuscript. 

Authors’ response: Thank you for pointing this out. As indicated in the comment (reviewers 1 and 3), we have revised this section of the manuscript to make it more concise as per the suggestions of the reviewers.

Comment 4: I suggest moving the description of the intervention to supplementary material and including only a brief summary of it in the manuscript. I also think it should be placed before talking about the outcomes. 

Authors’ response: Thank you for this comment, too. We have carefully revised this section to improve the readability of the manuscript as per your comment.

Comment 5: Page 14 paragraph 2 talks about data collection and then goes back to randomization, this is a little confusing. I would suggest authors to mention everything that has to do with randomization in the appropriate paragraph and not coming both.

Authors’ response: Thank you for your vital comment and for fixing our mistake. Now we have revised it based on your suggestion.

Comment 6: While the authors offer very detailed information and justifications for their model selection, I’m a little confused if their multilevel models were linear or Poisson models. It is not the standard to estimate ICC from a Poisson model and usually variance portioning is used. If fitting a logistic model then MOR is what is usually estimated as opposed to ICC If authors estimated ICC from non-linear models I would suggest that they add the formula of how they did it to the manuscript.

Authors’ response: Dear reviewer, thank you for your vital comment. As you correctly pointed out, it is not the standard and possible to estimate ICC from a Poisson model. However, we have provided a clear description of ICC estimate calculation using a logistic regression model in the data analysis and result section of this manuscript. To the best of our knowledge, it is possible to calculate the ICC estimate using a logistic regression model to determine whether or not a multilevel analysis is required. Besides, we have provided a stata do file that shows the formula for ICC estimation using stata software as per your request.

Comment 7: Why did authors decide the cutoff of p < 0.25 to be included in multivariate models.

Authors’ response: Thank you again. You raised a very critical point, but we haven’t only used the p-value cutoff point to screen candidate variables for multivariable analysis. A p-value of less than 0.25 and other variables of known clinical and social significance were included for additional multivariable analysis for this study. Utilizing a cutoff value of 0.25 is supported by most statistics books, statisticians, and literatures. We have decided to use the cutoff of p < 0.25 and other variables of known clinical and social importance to include candidate variables into multivariate models because most statistics books, literatures, and statisticians recommend by rule of thumb. Dear reviewer, kindly see the following references:

1. Bendel RB, Afifi AA. Comparison of stopping rules in forward “stepwise” regression. Journal of the American Statistical association. 1977 Mar 1;72(357):46-53. 

2. Mickey RM, Greenland S. The impact of confounder selection criteria on effect estimation. American journal of epidemiology. 1989 Jan 1;129(1):125-37.

3. Model building strategy for logistic regression: purposeful selection available from https://atm.amegroups.org/article/view/9400/pdf

Comment 8: What was the reasoning behind the effect modification tests? Which variables were tested? I would suggest authors to elaborate more on this.

Authors’ response: Thank you too for this comment. The reason we conducted the effect modification tests was to assess variables that modify the effect of our intervention. We have already provided the variables that were tested for effect modification and every detail of effect modifiers in supplementary file 1. 

Comment 9: In general, authors provide many statistical terms and details to the methods section that makes it hard to follow. I would suggest authors simplify it and add only the important information readers would need in order to replicate this analysis. The section is very hard to follow and all the justification and details can be distracting.

Authors’ response: Thank you a lot for this appreciated comment. We have accepted the suggestion and done the needed revision. Now this section has been revised as per your suggestion.

Comment 10: On the description of table 1, it would be important to highlight the differences between groups, given that this is an RCT and Table 1 should be used to assess balance between groups. The statistically significant differences for Mass Media access and wealth index are important and could be associated with the outcome as well so should be highlighted

Authors’ response: Thank you for this remark. We have highlighted all variables that significantly differ between two arms during bivariable analysis. These variables were considered as potential confounders and controlled using multivariable analysis.

Comment 11: Similar to previous comment, it is important to highlight significant differences between groups, previous history of neonatal death and ODS during last pregnancy could be associated with the outcomes as well and should be highlighted as potential confounders.

Authors’ response: Thank you for this remark, too. As indicated in above comment 10, these variables were considered potential confounders and controlled using multivariable analysis.

Comment 12: I find tables 3 and 4 a little confusing. I would separate the N(%) of good and poor knowledge into a separate table and just leave the model results. Also, it is not very clear which models the authors are showing. In the methods they refer to a sequence of four models and they only show results for 2, clarifying that would be very helpful.

Authors’ response: Thank you for the likely comment. This comment is also shared by reviewer 4; we have accepted comment and revised tables 3 and 4 as per suggestions of reviewers. We have provided the results of the final model (model 4) in this manuscript, but not for two models. The final model that contains both individual and community-level determinants was used to present the findings of this manuscript because it is the best fitted data as compared to the other three consecutive models (models 0, 1, and 2).

Comment 13: While the information provided is good and authors do a good job of comparing their findings to other literature, the limitations section is very long and disorganized. I would strongly advise to summarize better the limitations with regards to clustered RCTs and then they can talk about limitations of analysis or other types instead of going back and forth which makes it hard to follow. 

Authors’ response: Thank you for this comment. We have accepted your comment and did the required revisions as per your comment to make the section more concise.

Reviewer 3

Comment 1: Amanuel Yoseph and co-authors evaluated the effect of community-based health education intervention facilitated by women's groups (women's development team) on mothers' knowledge of SDG and BPCR practice in Sidama region in the southern Ethiopia. I would like to thank the authors for this excellent manuscript. A problem that affects a high proportion of women in underdeveloped countries is outlined: high maternal morbidity and mortality. But, even more importantly, the authors present evidence on effective strategies to mitigate this important public health problem. Adding to the relevance of the topic is the correct design, the careful handling of ethical aspects, a robust analysis and the writing of the manuscript.

Authors’ response: Thank you for your kind remark. 

Comment 2: My only suggestion is to review the title, its length seems a bit excessive to me.

Authors’ response: Thank you a lot for this valuable comment. This comment is also shared by reviewer 1; we have accepted the suggestion and made the required revision as per the suggestion of both reviewers. 

Comment 3: Additionally, I consider the comment "while another paper is focused on three skilled [...], in the "Study variables" section, unnecessary 

Authors’ response: Thank you for the genuine and plausible comment. We have removed this unnecessary description from the manuscript as per your suggestion. 

Comment 4: Congratulations, I really enjoyed reading the manuscript.

Authors’ response: Thank you very much again for your kind remark. 

Reviewer 4

Comment 1: Introduction Section: From our perspective, the ideas in this section are long and repetitive. We recommend condensing sentences 2, 3, and 8. Sentence 6 could be rewritten or omitted.

Authors’ response: Thank you for this comment. We have accepted your comment and did the required revisions as per your comment to make the introduction section more concise. 

Comment 2: In order to develop an educational intervention that addresses complications and reduces maternal and infant mortality rate, it is crucial to have a solid theoretical and causal framework. The causal framework is a diagram that considers the factors associated with social knowledge and establishes clear theoretical relationships between these factors. By incorporating a causal framework, the intervention can show its function and impact, highlighting the desired variable of change and the desired health outcomes. In addition, the baseline measure allows identifying should identify the variables that should be included in the analysis conceptual and ensure factor similarity between the intervention and non-intervention groups. It also allows for including relevant variables in the statistical analysis. The causal framework will justify the importance and scope of the intervention. For further help, see the following references:

Authors’ response: Thank you for the important comment. This study was developed based on the theoretical framework, which was well elaborated in another publication of this project. We have already provided all details of the theoretical and causal framework as per your suggestion in another publication. It was denied from this manuscript to avoid plagiarism of similar theoretical frameworks and figures from that paper. Dear reviewer, kindly see this link to the publication: https://doi.org/10.3390/healthcare12101045.

Comment 3: It is recommended that the participant recruitment process be described in more detail both in the text and in Figure 1.

Authors’ response: Thank you for this comment. Our manuscript participant recruitment process seems to contain detailed information. However, we have revised this section based on your suggestion to make it clearer for the reader.

Comment 4: I am confused by the way participant recruitment is described. If pregnant women were selected by visiting all households, if so, how was the response rate assessed if some households had a pregnant woman who did not want to be interviewed, or was not home because she had been hospitalized for complications of pregnancy? 

Authors’ response: Dear reviewer, thank you for your vital comment. As you correctly pointed out, there is a possibility of refusal of study participants due to different reasons. However, we haven’t practical experienced the concern you raised because we provided detail information about the aims and significance of the study, method of selection, benefits, and harms of the study to pregnant women. Due to this, study participants actively participated in this study during data collection. Besides, our data collectors visited a single pregnant home three times to avoid non-response due to absenteeism. If women were absent from home after three consecutive visits, they were considered non-respondent. We had never experienced hospitalized women during data collection because our project data was collected after 45 days of childbirth. We only experienced non-response due to change of place, abortion, stillbirths, and mortality during our data collection, as clearly shown in Figure 1.

Comment 5: Other questions: How many households were visited in total?

Authors’ response: Thanks for this query. However, we have clearly provided the information in the respondent details part of the result. Between November and December 2022, we assessed 1,440 pregnant mothers or households for eligibility; 1,126 from 24 clusters satisfied the criteria and were recruited for the study.

Comment 6: What is the rate of refusal or nonparticipation? 

Authors’ response: Thanks for this query, too. A total of 1,070 (95.02%) women were available for outcome assessment during the data collection period: 540 in the intervention (94.91%) and 530 in the control (94.13%). The study's overall response rate was 95.02%. The proportion of women lost to follow-up was comparable among both groups (4.98% in the intervention group vs. 5.87% in the control group).

Comment 7: During what stage of the recruitment process must women provide consent to take part? What was the community consent process? Given that the intervention was part of a major project, at what point in the study process were the groups randomized? 

Authors’ response: Thanks for this comment. The written informed consent was received from all pregnant women who met inclusion criteria before randomization and enrollment. The community leaders approved on behalf of the community before recruitment and randomization. Randomization was conducted after successful screening, consent, and recruitment of study participants.

Comment 8: How do you measure the equivalence of the two groups in personal, social, and health service variables that theoretically take part in knowledge? 

---

## [Editor Report · Decision Letter 2]

4 Oct 2024

Effect of community health education on mothers' knowledge of obstetric danger signs and birth preparedness and complication readiness practices in southern Ethiopia: A cluster-randomized controlled trial

PONE-D-23-40070R2

Dear Dr. Amanuel,

We’re pleased to inform you that your manuscript has been judged scientifically suitable for publication and will be formally accepted for publication once it meets all outstanding technical requirements.

Kind regards,

Doris Verónica Ortega-Altamirano, PhD

Academic Editor

PLOS ONE

Additional Editor Comments (optional):

The manuscript is ready for publication
---

## [Editor Report · Acceptance letter]

18 Oct 2024

PONE-D-23-40070R2 

PLOS ONE

Dear Dr. Yoseph , 

I'm pleased to inform you that your manuscript has been deemed suitable for publication in PLOS ONE. Congratulations! Your manuscript is now being handed over to our production team.

Kind regards, 

on behalf of

Dr. Doris Verónica Ortega-Altamirano 

Academic Editor

PLOS ONE